# Outcome and Prognostic Factors of COVID-19 Infection in Swiss Cancer Patients: Final Results of SAKK 80/20 (CaSA)

**DOI:** 10.3390/cancers14092191

**Published:** 2022-04-27

**Authors:** Markus Joerger, Yannis Metaxas, Khalil Zaman, Olivier Michielin, Nicolas Mach, Adrienne Bettini, Andreas M. Schmitt, Nathan Cantoni, Clemens B. Caspar, Sonja Stettler, Roma Malval, Miklos Pless, Christian Britschgi, Christoph Renner, Dieter Koeberle, Jessica D. Schulz, Christoph Kopp, Stefanie Hayoz, Anastasios Stathis, Roger von Moos

**Affiliations:** 1Department of Oncology, Cantonal Hospital, 9007 St. Gallen, Switzerland; 2Department of Medical Oncology, Cantonal Hospital, 8596 Muensterlingen, Switzerland; ioannis.metaxas@stgag.ch; 3Breast Center, Department of Medical Oncology, University Hospital, 1011 Lausanne, Switzerland; khalil.zaman@chuv.ch; 4Department of Medical Oncology, University Hospital, 1011 Lausanne, Switzerland; olivier.michielin@chuv.ch; 5Department of Medical Oncology, University Hospital, 1205 Geneva, Switzerland; nicolas.mach@hcuge.ch; 6Department of Medical Oncology, HFR Fribourg-Hôpital Cantonal, 1752 Fribourg, Switzerland; adrienne.bettini@h-fr.ch; 7Department of Medical Oncology, University Hospital, 4031 Basel, Switzerland; andreasmichael.schmitt@usb.ch; 8Oncology, Hematology and Transfusion Medicine, Cantonal Hospital, 5001 Aarau, Switzerland; nathan.cantoni@ksa.ch; 9Division of Hematology and Oncology, Cantonal Hospital, 5404 Baden, Switzerland; clemens.caspar@ksb.ch; 10Division of Medical Oncology, Lucerne Cantonal Hospital, 6000 Lucerne, Switzerland; sonja.stettler@luks.ch; 11Hôpital du Valais, 1951 Sion, Switzerland; roma.malval@hopitalvs.ch; 12Division of Medical Oncology and Hematology, Cantonal Hospital, 8400 Winterthur, Switzerland; miklos.pless@ksw.ch; 13Department of Medical Oncology and Hematology, University Hospital, 8091 Zurich, Switzerland; christian.britschgi@usz.ch; 14Department of Medical Oncology, Oncological Center Zurich, 8038 Zurich, Switzerland; christoph.renner@hirslanden.ch; 15Department of Oncology, Claraspital, 4058 Basel, Switzerland; dieter.koeberle@claraspital.ch; 16SAKK Coordinating Center, 3008 Bern, Switzerland; jessica.schulz@merck.com (J.D.S.); c.kopp@bluewin.ch (C.K.); stefanie.hayoz@sakk.ch (S.H.); 17Oncology Institute of Southern Switzerland, 6500 Bellinzona, Switzerland; anastasios.stathis@eoc.ch; 18Department of Medical Oncology/Hematology, Cantonal Hospital Graubuenden, 7000 Chur, Switzerland; roger.vonmoos@ksgr.ch

**Keywords:** COVID-19, coronavirus, pandemic, cancer, cancer treatment

## Abstract

**Simple Summary:**

The characterization of clinical outcomes and prognostic factors in COVID-19-infected cancer patients is important for health care authorities, policy makers and oncologists. We collected extensive data on COVID-19-infected cancer patients from 1 March 2020 over a one-year period. Among 455 patients included, death from COVID-19 infection occurred in 98 patients, resulting in a mortality rate of 21.5%. Age ≥ 65 years, non-curative disease, intensive care unit admission and oxygen requirement were particular and independent negative prognostic factors for death from COVID-19. COVID-19 severity and mortality in cancer patients remained high in a country (Switzerland) with a decentralized, high-quality, universal-acess health care system.

**Abstract:**

Purpose: These are the final results of a national registry on cancer patients with COVID-19 in Switzerland. Methods: We collected data on symptomatic COVID-19-infected cancer patients from 23 Swiss sites over a one-year period starting on 1 March 2020. The main objective was to assess the outcome (i.e., mortality, rate of hospitalization, ICU admission) of COVID-19 infection in cancer patients; the main secondary objective was to define prognostic factors. Results: From 455 patients included, 205 patients (45%) had non-curative disease, 241 patients (53%) were hospitalized for COVID-19, 213 (47%) required oxygen, 43 (9%) invasive ventilation and 62 (14%) were admitted to the ICU. Death from COVID-19 infection occurred in 98 patients, resulting in a mortality rate of 21.5%. Age ≥65 years versus <65 years (OR 3.14, *p* = 0.003), non-curative versus curative disease (OR 2.42, *p* = 0.012), ICU admission (OR 4.45, *p* < 0.001) and oxygen requirement (OR 20.28, *p* < 0.001) were independently associated with increased mortality. Conclusions: We confirmed high COVID-19 severity and mortality in real-world cancer patients during the first and second wave of the pandemic in a country with a decentralized, high-quality, universal-access health care system. COVID-19-associated mortality was particularly high for those of older age in a non-curative disease setting, requiring oxygen or ICU care.

## 1. Introduction

From a local respiratory disease epidemic within the Chinese province of Wuhan, severe acute respiratory syndrome coronavirus 2 (SARS-CoV-2) and the resultant coronavirus disease 2019 (COVID-19) rapidly emerged as a pandemic that brought health care systems to the brink of collapse in many regions of the world [1]. Early on in the pandemic, it became clear that the risk of morbidity and mortality from COVID-19 was not uniform across infected patients but was rather dependent on specific patient characteristics, including age and comorbidities, such as immunosuppression and pulmonary and malignant comorbidities, among others [2,3,4,5]. Over time, accumulating data suggested an increased risk of high morbidity and mortality from COVID-19 in cancer patients [3,6,7,8,9,10,11,12,13,14]. In the three largest non-cancer type-specific registry trials during the initial phase of the pandemic, COVID-19-associated mortality was 14% among 4966 cancer patients from the COVID-19 and Cancer Consortium (CCC19) study [11] (13% in the initial report in 928 cancer patients [6]), 10.9% among 1794 cancer patients in the US National Veterans Affairs Study [10] and 32.3% among 351 cancer patients in the Dutch Oncology COVID-19 consortium study [9]. Important cancer type-specific registry trials include the TERAVOLT initiative, which found an overall COVID-19-associated mortality of 33% among 200 patients with thoracic cancers [14], 33% among 697 patients with hematological malignancies [12] and 21% among 228 prostate cancer patients [15]. Accordingly, clinicians involved in cancer care are facing competing risks from cancer progression and COVID-19. These include issues such as potentially increased risk of COVID-19 infection with hospital visits for cancer care or more severe course of COVID-19 infection with concurrent anticancer treatment. This data vacuum led to frequent adjustments of cancer patient management during the COVID-19 pandemic, even in regions with low COVID-19 incidence [16,17,18]. Data and recommendations to inform the cancer care community about how to handle these difficult issues in everyday practice are critical but still usually limited to specific tumor entities [19,20,21,22]. To build such recommendations, more data on the outcomes as well as important prognostic factors in the more general cancer patient population susceptible to COVID-19 infections are crucial. Switzerland was a country heavily affected by COVID-19 in the first two waves, accumulating roughly 1 million confirmed cases (12% of the population) and 11,500 deaths from COVID-19 (0.13% of the population, updated 5 December 2021). Thus, greater knowledge of local data regarding the COVID-19 outbreak is of utmost importance for the management of patients with cancer.

We describe clinical outcomes in symptomatic COVID-19-infected patients with solid or hematological malignancies in secondary and tertiary cancer care sites covering the majority of the Swiss population in a country with a distinctly decentralized health care system.

## 2. Methods

### 2.1. Study Design and Eligility Criteria

SAKK 80/20 was executed by the Swiss Group for Clinical Cancer Research, which is a nationwide academic consortium of oncology physicians (hematologists, medical oncologists, radio-oncologists, surgeons) in Switzerland. The study was an observational study with a cross-sectional design, allowing the obtainment of real-life information on SARS-CoV-2 infection in cancer patients and its possible impact on outcomes of individual cancer disease. The study aimed to assess all patients with histologically or cytologically confirmed malignant tumors who had been additionally diagnosed with SARS-CoV-2 infection during a time period of 12 months. This allowed us to analyze the data from the SARS-CoV-2 2020 summer peak and the 2020/2021 winter peak. SAKK 80/20 was designed as a national registry to support rapid implementation of data on COVID-19 infection in cancer patients in oncological practice. SAKK 80/20 collected extensive clinical and outcome data from symptomatic SARS-CoV-2-infected, adult (≥18 years of age) cancer patients between 1 March 2020 and 15 March 2021. All patients have or had histologically or cytologically confirmed early or advanced solid or hematological malignancies independent of their specific treatment setting, including patients with a history of cancer. SARS-CoV-2 infection was either confirmed by polymerase chain reaction (PCR)-based nasopharyngeal swab testing or diagnosed by clinical and radiological findings in a few cases. Testing for SARS-CoV-2 followed the guidelines set out by the national authorities, which can be found with the following link: https://www.bag.admin.ch/bag/de/home/krankheiten/infektionskrankheiten-bekaempfen/meldesysteme-infektionskrankheiten/meldepflichtige-ik/meldeformulare.html (accessed on 27 March 2022). Twenty-three Swiss secondary and tertiary care sites participated in SAKK 80/20, representing almost all study sites of the Swiss Clinical Cancer Research (SAKK) network and covering an estimate of more than two-thirds of the Swiss population. The SAKK research network does not include oncologists in private practice and potential limitations are outlined in the discussion section. The study collected baseline clinical data, including patient characteristics, prior and current anticancer treatment, comorbidities, concomitant medications, symptoms of SARS-CoV-2 infection, the type of SARS-CoV-2 diagnosis (nasopharyngeal swab versus clinico-radiological diagnosis), specific treatment of SARS-CoV-2 infection and clinical outcome until the end of the study period, this being 15 March 2021. Subsequently, the data provided were centrally entered into an electronic clinical record form (eCRF) using a secured digital database (SecuTrial). The study was approved by the responsible Ethics Committee (EKOS, No. 2020-00804) and processed data were fully pseudonymized. Patients gave their written informed consent for trial participation.

### 2.2. Study Objectives and Statistical Analysis

The primary objective of SAKK 80/20 was to collect clinical data and investigate the outcomes of cancer patients with symptomatic SARS-CoV-2 infection. The secondary objective of the study was to define prognostic factors in this population with acquired SARS-CoV-2. Co-primary endpoints included the proportion of complicated SARS-CoV-2 infections, i.e., the need for hospitalization and admission to the intensive care unit (ICU), and the proportion of fatal cases due to SARS-CoV-2 infection. Secondary endpoints included overall survival (OS) and the proportion of patients requiring mechanical ventilation. The following parameters were considered as potential prognostic covariables: patient age, gender, tumor disease (solid versus hematological malignancy, the latter including lymphoma, multiple myeloma, leukemia and myelodysplastic syndrome), curative versus palliative disease setting, type of systemic anticancer treatment (chemotherapy, targeted anticancer treatment (including tyrosine kinase inhibitors and monoclonal antibodies, except immune checkpoint inhibitors), immunotherapy (i.e., immune checkpoint inhibitors), endocrine treatment), comorbidities (including cardiovascular and pulmonary disease, diabetes, obesity) and comedication. Cardiovascular comorbidities included heart valvular, myocardial and coronary disease, arterial hypertension, pulmonary embolism, pulmonary heart disease, endocarditis, arrhythmias, cerebrovascular disease, including cerebral hemorrhage, peripheral vascular disease, phlebitis and thrombophlebitis, among others. Pulmonary comorbidities included bronchitis, emphysema, asthma, chronic obstructive pulmonary disease, pleura effusions, among others.

The characteristics of patients with resolved versus fatal cases of SARS-CoV-2 infection were analysed. Descriptive statistics were used for baseline characteristics. Standard tests were used to check univariable associations between categorical and categorical (Fisher’s exact test) or categorical and continuous (Wilcoxon rank-sum test) variables. Additional categorical testing was performed for continous variables such as age. Logistic regression was used to test multivariable associations between binary outcomes and other patient characteristics. We did not implement correction for multiple testing due to the exploratory and hypothesis-generating nature of the study. All analyses were performed using SAS version 9.4. and R version 3.6.0 (www.r-project.org (accessed on 5 March 2020)) or more recent versions thereof. All statistical tests were performed two-sided.

## 3. Results

### 3.1. Patient Population

With a cutoff date of 15 March 2021, 455 consenting patients were included between 1 March 2020 and 15 March 2021 and proceeded to the final analysis. Detailed baseline characteristics are presented in Table 1. Median age was 69 years (range 27 to 95) and 261 (57.4%) patients were male. SARS-CoV-2 infection was diagnosed based on PCR-based nasopharyngeal swabs (PCR+) in 428 (94.1%) patients and based on clinico-radiological findings in 27 (5.9%) patients. The most frequent malignancies were breast cancer in 81 (18%) cases, lung cancer in 57 (13%), prostate cancer in 37 (8%) and multiple myeloma in 21 (5%) cases; 205 (45%) patients had a malignancy in the non-curative setting. Systemic treatment at the moment of COVID-19 diagnosis or within 3 months prior to COVID-19 diagnosis included chemotherapy in 101 (22.5%) cases, targeted therapy in 94 (20.9%), steroids in 52 (11.6%) and checkpoint inhibitors in 33 (7.3%) cases; 158 (70.2%) out of those 225 patients were on active systemic anticancer treatment at the time of diagnosis of COVID-19. Radiotherapy within 3 months prior to COVID-19 diagnosis was received by 39 (8.7%) patients; surgery was received by another 39 (8.7%) patients. Besides diagnosis of solid or hematological malignancy, 378 (83.1%) patients had one or more relevant comorbidities, most frequently cardiovascular disease in 244 (53.6%) patients; 166 (36.5%) patients were current or former smokers. The three most frequent symptoms at the time of COVID-19 diagnosis included fever in 286 (62.9%) patients, cough in 265 (58.2%) patients and fatigue in 180 (39.6%) patients.

### 3.2. COVID-19 Treatment and Coutcome

COVID-19 clinical outomes are outlined in Table 2. In total, 98 (21.5%) patients died from COVID-19. Cancer patients admitted to the ICU had the highest mortality, with 35 (56.5%) patients dying from COVID-19. Out of the 98 patients who died from COVID-19, the majority (63 (64.3%)) of patients were not admitted to the ICU. Overall, 285 (62.7%) patients were either admitted to the hospital for COVID-19 infection (241 patients, 53.0%) or were already hospitalized (44 patients, 9.7%). Treatment of hospitalized patients included oxygen therapy in 213 (74.7%) patients, non-invasive ventilation in 45 (15.8%) patients, invasive ventilation in 43 (15.1%) patients and ICU admission in 62 (21.8%) patients. The most frequent systemic treatment due to SARS-CoV-2 infection included antibiotics in 227 (49.9%) patients, with further treatments as outlined in Table 2. Time to resolution of symtoms from COVID-19 infection was a median of 19 days (interquartile range: 13 to 30 days; range: 1 to 225 days). Anti-SARS-CoV-2 systemic therapies had no significant impact on clinical outcome. Median time of hospitalization was 12 days (interquartile range: 7 to 21 days; range: 1 to 139 days).

### 3.3. COVID-19 Prognostic Factors in Cancer Patients

Eight significant negative prognostic factors for survival from COVID-19 were identified using univariable logistic regression modeling, including male gender (HR 1.93, 95% CI 1.19–3.10, *p* = 0.007), age ≥ 65 years (HR 4.45, 95% CI 2.51–7.91, *p* < 0.001), non-curative setting (HR 2.03, 95% CI 1.23–3.33, *p* = 0.005), cardiovascular disease (HR 2.47, 95% CI 1.53–4.00, *p* < 0.001), ICU admission (HR 6.89, 95% CI 3.89–12.20, *p* < 0.001) and oxygen treatment (HR 38.80, 95% CI 13.92–108.13, *p* < 0.001). Importantly, neither the region of the patients who were treated (Southern versus Western versus Northern Switzerland), nor the type of malignancy (hematological versus solid tumors, HR = 1.18, 95% CI 0.72–1.95, *p* = 0.51), presence of metastatic disease (HR 1.27, 95% CI 0.71–2.25, *p* = 0.42), prior chemotherapy (HR 1.33, 95% CI 0.79–2.23, *p* = 0.28), prior immunotherapy (HR 1.55, 95% CI 0.71–3.36, *p* = 0.27) or lung comorbidities (HR 1.20, 95% 0.65–2.21, *p* = 0.65) were significantly associated with COVID-19-associated fatalities when submitted to univariable testing. The results of uni- and multivariable logistic regression modeling are outlined in Table 3. The three independent negative prognostic factors most strongly associated with COVID-19-associated fatalities from multivariable testing were oxygen requirement (HR 20.28, 95% CI 6.86–59.95, *p* < 0.001), ICU admission (HR 4.45, 95% CI 2.16–9.17, *p* < 0.001) and age ≥ 65 years (HR 3.14, 95% CI 1.49–6.64, *p* = 0.003). Chemotherapy within three months prior to COVID-19 diagnosis, patient gender and the type of malignancy (hematological versus solid tumors) were not significantly associated with COVID-19-associated fatalities with multivariable testing. Active systemic anticancer treatment at the time of diagnosis of COVID-19 was not significantly associated with mortality (HR = 0.60, 95% CI 0.35–1.00, *p* = 0.05) with univariable testing, nor was it forwarded to multivariable testing. The use of steroids within three months prior to the diagnosis of COVID-19 was marginally associated with mortality from COVID-19 (HR = 1.90, 95% CI 1.02–3.55, *p* = 0.044) with univariable testing, but this did not resist multivariable testing. Figure 1 outlines the course of COVID-19 in all study patients.

## 4. Discussion

The SAKK 80/20 (CaSa) registry was initiated in order to identify clinical characteristics of patients with early or advanced solid or hematological malignancies associated with an increased risk of fatal outcome from SARS-CoV-2 infection. We found higher age (modeled as age ≥ 65 years), ICU admission, oxygen requirement and non-curative disease setting to be significant, independent and clinically highly relevant negative prognostic factors in cancer patients with SARS-CoV-2 infections. This has to be seen against the background of a cancer population that has a substantially higher median age compared to the average Swiss population (69 years in SAKK 80/20 versus 42.7 years in the Swiss population), as well as a higher proportion of males versus females (57.4% in SAKK 80/20 versus 49.5% in the Swiss population). The median patient age in our study, however, is consistent with the median age of Swiss cancer patients (69.8 years for men, 68.1 years for women) according to the Swiss Federal Statistical Office (FOS) [23]. SAKK 80/20 is one of the few nation-wide COVID-19 registries and its particularities include extensive data collection of patient characteristics, including co-medication and co-morbidities, careful individual patient follow-up until recovery from SARS-CoV-2 infection or death, and study execution in a country with a highly decentralized health care system, different than countries, including the US, UK and the Netherlands [9]. Important characteristics of Switzerland’s health care systemic include three different levels of government (the federal level, the regional or ‘cantonal’ level and the municipalities for social services), recognized civil society organizations, such as associations of health insurers and health care providers, and the strong political impact of the Swiss citizen who can veto or demand reform through public referenda [24].

Higher age is the most consistent negative prognostic factor across clinial studies in cancer patients affected by COVID-19 [3,6,9,10,11,12], and our data confirm higher age as a significant, independent and clinically relevant risk factor for higher SARS-CoV-2-associated mortality. Male gender was identified as a negative prognostic factor in the two large US cancer cohorts [6,11] and by the Dutch Oncology COVID-19 consortium [9]. In SAKK 80/20, male gender was a significantly and negatively associated factor with mortality from SARS-CoV-2 infection in cancer patients (HR 1.93, 95% CI 1.19–3.10, *p* = 0.007), but this did not resist multivariable logistic regression analysis (HR 1.12, 95% CI 0.60–2.11, *p* = 0.722). It remains unclear whether co-linearity between gender and tumor entity (i.e., breast and lung cancer) may have impacted multivariable analysis in this regard. Chemotherapy prior to or at the time of SARS-CoV-2 diagnosis has previously been identified as a risk factor for mortality due to COVID-19 in cancer patients according to some registry studies [3,9,11,13]. Chemotherapy was not identified as a negative prognostic factor in the US cohort study of Jee et al. [25], the UK registry study by Lee et al. [26] or the US Veterans Affairs Study by Fillmore et al. [10]. According to the multivariable logistic regression model in SAKK 80/20, chemotherapy within 3 months prior to COVID-19 diagnosis was not significantly and independently associated with SARS-CoV-2-associated mortality. Other systemic anticancer treatment, particularly immunotherapy, was also not associated with mortality from SARS-CoV-2 infection, and this is similar to results from other registry studies [9,10,11,13]. However, the use of steroids within three months prior to the diagnosis of COVID-19 was a negative prognostic factor in the univariable model (HR = 1.90, 95% CI 1.02–3.55, *p* = 0.044), similar to what has been described by the Dutch Oncology COVID-19 consortium [9]. While dexamethasone decreases mortality from SARS-CoV-2 infection in patients requiring respiratory support [22], the use of steroids as part of anticancer treatment at the time of diagnosis of COVID-19 may represent an unfavourable prognostic factor but is unlikely to be a prominent negative prognostic factor. Hematological versus solid malignancy was identified as being unfavourably associated with mortality from SARS-CoV-2 infection in cancer patients in some studies [9,11,25], while we only found a weak numerical association between hematological malignancy and mortality from SARS-CoV-2 infection (HR 1.18, 95% CI 0.72–1.95, *p* = 0.512). In the publication of the Dutch Oncology COVID-19 consortium, hematological malignancy was a strong negative prognostic factor with regard to COVID-19-associated mortality (OR 2.15, 95% CI 1.30–3.57, *p* = 0.003), as it was in the COVID-19 and Cancer Consortium publication (OR 1.44, 95% CI 1.10–1.87) and the study by Jee et al. (HR 2.10, 95% CI 1.36–3.24, *p* < 0.01). In all of these studies, including SAKK 80/20, roughly one quarter of cases were hematological versus solid cancer cases. This is of particular interest given recent data on the limited efficacy of SARS-CoV-2 vaccines in patients with hematological malignancies [27].

SAKK 80/20 covers the first and second wave of symptomatic COVID-19 cases in Swiss cancer patients. While only very few national guidelines on the clinical management of COVID-19 (non-cancer) patients have been published so far [28], the Swiss Federal Office of Public Health (FOPH) has continually published national guidelines on SARS-CoV-2 testing, disease symptoms and treatment, measures of prevention and subsequent recommendations for SARS-CoV-2 vaccination [29]. The guidelines have changed over time and were the basis for SARS-CoV-2 testing and diagnosis as implemented in the SAKK 80/20 study. In October 2020, the FOPH issued a recommendation to weigh the risks and benefits of anticancer treatment regarding the current and changing situation of the COVID-19 pandemic. This recommendation included a warning to use aplastic chemotherapy only in the curative setting or in cases of expected substantial survival benefit and recommendations to limit the number of chemotherapy cycles to three to four in the non-curative setting, to generously use G-CSF and prophylactic antibiotics in specific cases and to limit the used of high-dose chemotherapy and (autologous/allogen) stem-cell transplantation. While there are no data on the adaptation of anticancer treatment in Switzerland during the COVID-19 pandemic, data from SAKK 80/20 support adequate systemic anticancer treatment, including chemotherapy, in well-monitored cancer patients, as our data do not suggest a detrimental effect from such treatment, including chemotherapy, within 3 months prior to COVID-19 diagnosis. A further consideration is the potential triage of patients at times of ICU resource scarcity, as was the case at the heights of the COVID-19 pandemic waves in Switzerland. The Swiss Academy of Medical Sciences (SAMW) issued such guidance in March 2020, with subsequent updates as the pandemic evolved [30]. These guidelines make a clear statement that comorbidities such as cancer must not be seen in themselves as a factor of prognostic relevance and hence individualized decisions have to be made with regard to intensive care in SARS-CoV-2-infected cancer patients. Again, results from SAKK 80/20 must not inform individual decision making in the cases of critically ill COVID-19 cancer patients but they can inform caregivers about prognostic conditions and support communication between caregivers and with the patients and their families. While any direct comparison of COVID-19 mortality in cancer versus non-cancer patients is methodologically not justifiable, we found COVID-19-associated mortality in cancer patients (21.5%) to be substantially higher than in the general Swiss population (1.06% as of 11 December 2021). Obviously, SARS-CoV-2 infections may be more readily detected in the general population where routine testing without symptoms is more frequent than in symptomatic cancer patients included in SAKK 80/20, resulting in an overestimation of COVID-19 mortality in cancer patients.

SAKK 80/20 has several limitations, including a potential selection bias due to including data provided by medical oncologists based at larger and middle-sized hospitals. This may have resulted in an overestimation of COVID-19-associated mortality, as these cancer institutions have an overrepresentation of patients with active or more complex cancer disease. Despite data collection by professional clinical trial coordinators and a robust data quality assurance system, data collection across multiple sites may result in selection biases, reporting errors and missing and and unknown data. Patients who had completed cancer treatment and were referred back to their general practitioner for follow-up visits, as well as cancer patients who were diagnosed with SARS-CoV-2 infection outside the hospital, may not have been registered in this national registry trial. Furthermore, SARS-CoV-2 testing policy in Switzerland changed over time [29], resulting in varying degrees of underestimation of the total number of SARS-CoV-2-infected cancer patients. Our study has various strenghts, including the prospective study design, coverage of the majority of the Swiss cancer population, extensive data collection and patient follow-up until recovery from SARS-CoV-2 infection or death. Importantly, despite being non-centralized, the Swiss health care system provides high-quality, universal access to medical care, and national guidelines provided homogeneity with regard to cancer management during the COVID-19 pandemic. Accordingly, the results of the SAKK 80/20 study are representative of the Swiss real-world cancer patient population.

## 5. Conclusions

We confirmed high COVID-19 severity and a mortality rate of 21.5% in real-world cancer patients during the first two waves of the pandemic in a country with a decentralized, high-quality, universal-access health care system. COVID-19-associated mortality was particularly high among those of older age, in a non-curative disease setting, requiring oxygen or ICU care. The rates of hospitalization and ICU admission for COVID-19 in cancer are substantial, revealing the urgency of avoiding exposure and prioritizing vaccination. SAKK 80/20 does not suggest that cancer patients receiving adequate systemic anticancer treatment, including chemotherapy, are at increased risk of COVID-19-associated mortality, indirectly supporting the delivery of adequate treatment to cancer patients, including potentially curative treatment, such as surgery and adjuvant chemo- or radiotherapy.

## Figures and Tables

**Figure 1 cancers-14-02191-f001:**
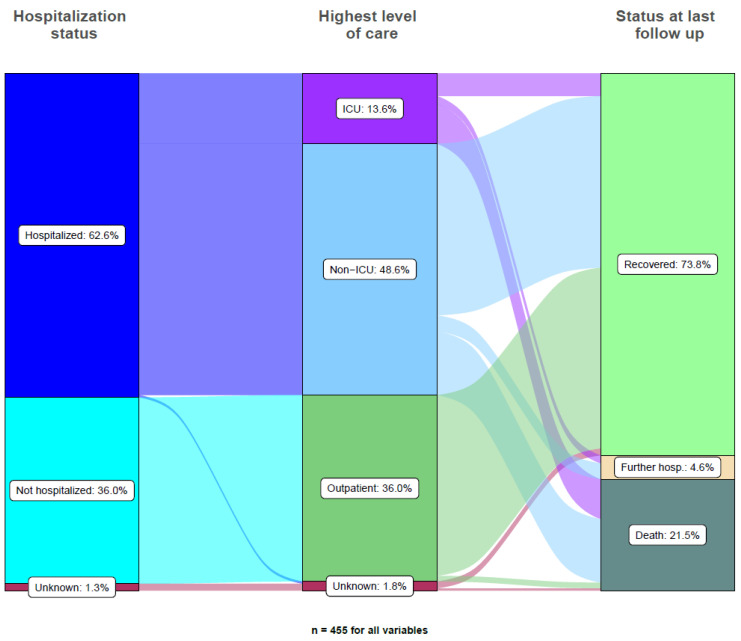
Alluvial plot of hospitalization and clinical courses for all 455 cancer patients.

**Table 1 cancers-14-02191-t001:** Patient demographics and clinical characteristics.

Patient Characteristics	N	%
Gender		
male	261	57.4
female	194	42.6
Age		
65+	273	60.0
<65	182	40.0
Geographic region		
Western CH	195	42.9
Northern CH	199	43.7
Southern CH	61	13.4
Tumor entity		
solid tumor	332	73.7
hematological malignancy	119	26.3
breast cancer	81	17.8
lung cancer	57	12.5
prostate cancer	37	8.1
multiple myeloma	21	4.6
Prognostic tumor setting		
non-curative	205	50.6
curative	200	49.4
Systemic anticancer treatment		
yes	225	49.2
no	224	50.8
chemotherapy	101	22.5
targeted anticancer agents	94	20.9
steroids	52	11.6
endocrine treatment	43	9.6
immunotherapy *	34	7.6
Comorbidity other than COVID-19		
yes	378	83.1
no	77	16.9
cardiovascular disease	244	53.6
pulmonary disease	66	14.5
diabetes mellitus	63	13.8
adipositas	45	9.9
cachexia/malnutrition	34	7.5

Including tyrosine kinase inhibitors, monoclonal antibodies except immune checkpoint inhibitors, * immune checkpoint inhibitors, Abbreviations: CH = Switzerland; N = number of patients.

**Table 2 cancers-14-02191-t002:** SARS-CoV-2 infection outcome in cancer patients.

Patient Characteristics	N	%
Hospitalization for COVID-19		
yes	241	53.0
no	170	37.3
already hospitalized	44	9.7
Oxygen requirement		
yes	213	46.8
no	242	53.2
ICU admission		
yes	62	13.6
no	393	86.4
Invasive ventilation		
yes	43	9.5
no	412	90.5
COVID-19 associated death		
in all studied cancer patients	98	21.5
in hospitalized cancer patients	91	31.9
in cancer patients requiring oxygen	88	41.3
in cancer patients admitted to the ICU	35	56.5
Systemic treatment for COVID-19		
antibiotics	227	49.9
chloroquin	102	22.4
antivirals	61	13.4
steroids	78	17.1
fungistatics	41	9
tocilizumab	6	1.3

**Table 3 cancers-14-02191-t003:** Univariable and multivariable Cox regression models on COVID-19 associated death (outcome = death (yes/no), N = 431 complete observations, 93 deaths).

	Univariable			Multivariable		
	HR	95% CI	P	HR	95% CI	P
Gender						
male vs. female	1.93	1.19–3.10	0.007	1.12	0.60–2.11	0.722
Age						
65+ vs. <64	4.45	2.51–7.91	<0.001	3.14	1.49–6.64	0.003
Geographic region						
Western vs. Northern CH	0.82	0.51–1.35	0.442			
Southern vs. Nothern CH	1.36	0.71–2.61	0.354			
Tumor type						
hematological vs. solid malignancy	1.18	0.72–1.95	0.512	1.02	0.53–1.97	0.952
Prognostic tumor setting						
non-curative vs. Curative	2.03	1.23–3.33	0.005	2.42	1.22–4.83	0.012
Presence of metastases						
yes vs. no	1.27	0.71–2.25	0.422			
Chemotherapy						
yes vs. no	1.33	0.79–2.23	0.280	1.44	0.71–2.93	0.311
Immunotherapy *						
yes vs. no	1.55	0.71–3.36	0.269	3.15	0.95–10.38	0.06
Steroids						
yes vs. no	1.90	1.02–3.55	0.044	0.90	0.42–1.90	0.774
Anticancer treatment at the time of C19 diagnosis						
yes vs. no	0.60	0.35–1.00	0.050			
Comorbidity						
cardiovascular disease	2.47	1.53–4.00	<0.001	1.15	0.61–2.18	0.665
pulmonary disease	1.20	0.65–2.21	0.564	1.01	0.46–2.22	0.99
Oxygen requirement						
yes vs. no	38.80	13.92–108.13	<0.001	20.28	6.86–59.95	<0.001
ICU admission						
yes vs. no	6.89	3.89–12.20	<0.001	4.45	2.16–9.17	<0.001
Invasive ventilation						
yes vs. no	6.53	3.38–12.61	<0.001			

Abbreviations: HR = hazard ratio, CI = confidence interval, CH = Switzerland, ICU = intensive care unit; * immune checkpoint inhibitors.

## Data Availability

The data presented in this study are available on request from the corresponding author.

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
