# Peer review of "Outcome and Prognostic Factors of COVID-19 Infection in Swiss Cancer Patients: Final Results of SAKK 80/20 (CaSA)"

_cancers, 2022, doi:10.3390/cancers14092191_

Round 1
Reviewer 1 Report
The current observational cross-sectional study evaluated cancer patients' outcomes with symptomatic SARS-CoV-2 infection and defined prognostic factors using the SAKK 80/20 (CaSa) national registry of Switzerland, which has a decentralized, high-quality, universal access health care system.
Fever, cough, and fatigue were the most prevalent symptoms during the Covid19 diagnosis, with a reported covid19 associated mortality (21.5%) in cancer patients and the highest mortality being reported in ICU admitted cancer patients. The study identified factors that affected survival in COVID-19 as being male gender, cardiovascular disease, non-curative setting, age ≥65 years, ICU admission, and oxygen treatment in which the last four factors were clinically relevant prognostic factors for cancer patients with Covid19. Interestingly type of malignancy (hematological vs. solid tumors), presence of metastatic disease, prior chemo/immune therapy, or lung comorbidities were not significantly associated with COVID-19 fatalities.
The manuscript is clear, scientifically sound, well-structured, and very well related to the current scope of the journal.
The conclusions are consistent with the article's hypothesis, and the results
All the articles cited are from the last five years. The article does not have self-citations.
Author Response
Reviewer 1:
The current observational cross-sectional study evaluated cancer patients' outcomes with symptomatic SARS-CoV-2 infection and defined prognostic factors using the SAKK 80/20 (CaSa) national registry of Switzerland, which has a decentralized, high-quality, universal access health care system.
Fever, cough, and fatigue were the most prevalent symptoms during the Covid19 diagnosis, with a reported covid19 associated mortality (21.5%) in cancer patients and the highest mortality being reported in ICU admitted cancer patients. The study identified factors that affected survival in COVID-19 as being male gender, cardiovascular disease, non-curative setting, age ≥65 years, ICU admission, and oxygen treatment in which the last four factors were clinically relevant prognostic factors for cancer patients with Covid19. Interestingly type of malignancy (hematological vs. solid tumors), presence of metastatic disease, prior chemo/immune therapy, or lung comorbidities were not significantly associated with COVID-19 fatalities.
The manuscript is clear, scientifically sound, well-structured, and very well related to the current scope of the journal.
The conclusions are consistent with the article's hypothesis, and the results
All the articles cited are from the last five years. The article does not have self-citations.
Answer: We thank Reviewer 1 for thourough review.
Reviewer 2 Report
This is a study of a Swiss national registry of COVID-19 among patients with cancer with co-primary endpoints of assessing the proportion of complicated COVID infections and the mortality rate of the patients in the registry. My comments are below and organized by section:
Introduction: Should discuss evidence of higher COVID incidence and mortality among people with cancer in more detail - right now it is very vague. Would specifically mention other data on mortality rates from other registries e.g. CCC19.
Methods - This section needs more elaboration as certain details will be important to interpretation of the results.
- How representative is this population of people with cancer who got COVID (or people with cancer) in Switzerland?
- How was the cohort selected?
- The article frequently refers to Switzerland's distinctly decentralized healthcare system. If this is to be a unique feature of the study, it needs to be discussed in more detail. For example, the American health care system is completely decentralized. How is the Swiss system different, and how does this influence our understanding of the study findings?
- Did patients need to have active cancer to be included or was a history of cancer permitted?
- How many patients were asymptomatic?
- How were cardiovascular disease and "lung comorbidities" defined? Did simple hypertension count, for example?
- What characteristics necessitated ICU transfer? This varies greatly depending on the national (and often regional) context.
Results
- Figure 1 does not add any new information; the symptomatology of COVID is clear and I would remove this
- Lines 202-212 - Please clarify that this is the MV analysis
Discussion
- State more clearly how the median age of the cohort may have biased the sample
- Why do you think you did not see an increased risk among patients with hematologic malignancies when this has been seen in most other registries?
-
How do you explain no increased risk for metastatic disease (univariate) but increased risk for noncurative setting? (univariate/MV)
- What were the effects of systemic therapies on outcome?
- Please be careful in saying things were "numerically associated" when they were not statistically significant. This is not meaningful (e.g. lines 246-247, 258)
- Take caution in cherry picking results from both the UV and MV models, e.g. steroid use.
Author Response
Reviewer 2:
This is a study of a Swiss national registry of COVID-19 among patients with cancer with co-primary endpoints of assessing the proportion of complicated COVID infections and the mortality rate of the patients in the registry. My comments are below and organized by section:
Introduction: Should discuss evidence of higher COVID incidence and mortality among people with cancer in more detail - right now it is very vague. Would specifically mention other data on mortality rates from other registries e.g. CCC19.
Answer: We fully agree with the Reviewer and added more details on COVID-19 associated mortality in cancer patients to the Introduction Section.
Methods: This section needs more elaboration as certain details will be important to interpretation of the results.
How representative is this population of people with cancer who got COVID (or people with cancer) in Switzerland?
Answer: We agree that this is an important point. We added an estimate on the cancer population coverage of our trial in the Methods Section. Additionally, we outlined the potential bias that patient selection might have on the results of our trial in the Discussion Section (potential selection bias was already mentioned as a limitation of SAKK 80/20).
How was the cohort selected?
Answer: We added some more information on the participating cancer institutions / research network that allows the reader to appreciate the limitations of SAKK 80/20.
The article frequently refers to Switzerland's distinctly decentralized healthcare system. If this is to be a unique feature of the study, it needs to be discussed in more detail. For example, the American health care system is completely decentralized. How is the Swiss system different, and how does this influence our understanding of the study findings?
Answer: We added features of Switzerland’s decentralized health care system in the Discussion Section.
Did patients need to have active cancer to be included or was a history of cancer permitted?
Answer: Patients with a history of cancer were eligible for the study. We clarified this in the Methods Section.
How many patients were asymptomatic?
Answer: With regards to COVID-19, all patients were symptomatic, as there were no recommendations for mass screening of asymptomatic Swiss citizens during the time period when the study was conducted. With regards to the underlying cancer disease, we did not assess specific individual symptoms, assuming that patients with ‘curative’ disease setting were asymptomatic with this respect, i.e. 49.4% (n=200) of all patients.
How were cardiovascular disease and "lung comorbidities" defined? Did simple hypertension count, for example?
Answer: This is an important issue indeed, and we added more details on the categorization of comorbidities in the Methods Section.
What characteristics necessitated ICU transfer? This varies greatly depending on the national (and often regional) context.
Answer: The Swiss Academy of Medical Sciences (SAMW) issued a guidance on “Intensive care medicine: triage under resource scarcity” in March 2020 with subsequent updates as the COVID-19 pandemic evolved (https://www.samw.ch/en/Ethics/Topics-A-to-Z/Intensive-care-medicine.html). We prefer to refer to this document as the recommendations changed somewhat over time, and the different documents can readily be accessed online to allow the reader detailed information on the recommended ICU triage conditions.
Results: Figure 1 does not add any new information; the symptomatology of COVID is clear and I would remove this.
Answer: Figure 1 was removed from the manuscript as suggested by Reviewer 2.
Lines 202-212 - Please clarify that this is the MV analysis.
Answer: In the Results Section, we first refer to results from the univariable analysis (lines 201-211), then we refer to results from the multivariable analysis (lines 212-2017). Both results from univariable and multivariable analysis are specifically outlined in Table 3. We agree with Reviewer 2 that the text does not always clearly refer to the type of testing (univariable/multivariable). We amended accordingly.
Discussion:
State more clearly how the median age of the cohort may have biased the sample
Answer: The median age of the study cohort (69 years) is consistent with the median age of the Swiss cancer patient (69.8 years for men, 68.1 years for women) according to the Swiss Federal Staticial Office (FOS). We added this statement and the respective reference.
Why do you think you did not see an increased risk among patients with hematologic malignancies when this has been seen in most other registries?
Answer: We did see that patients with hematological malignancies had an inferior clinical outcome of COVID-19 infection – i.e. 18% higher mortality – but this was not statistically significant. We had some intense internal discussions, but there seems not to be a very obvious explanation for the less pronounced negative impact of hematological malignancy on COVID-19 clinical outcome.
How do you explain no increased risk for metastatic disease (univariate) but increased risk for noncurative setting? (univariate/MV).
Answer: The presence of metastatic disease was associated with a 27% higher mortality from COVID-19 infection (which was not statistically significant). We think that the main issue is the considerable proportion of patients with hematological malignancy (26.3%, n=119) who are not categorized into ‘metastatic’ versus ‘non-metastatic’ disease.
What were the effects of systemic therapies on outcome?
Answer: With regards to anti-SARS-CoV-2 systemic therapies, no significant impact on clinical outcome was found. We added this information to the manuscript.
Please be careful in saying things were "numerically associated" when they were not statistically significant. This is not meaningful (e.g. lines 246-247, 258).
Answer: We acknowledge the issue raised by Reviewer 2 and amended accordingly.
Take caution in cherry picking results from both the UV and MV models, e.g. steroid use.
Reviewer 3 Report
The manuscript was prepared very well. I congratulate the authors for the preparation of the manuscript
However, I have the following comments:
Introduction
- Some references to the different types of cancer for which similar studies have been performed should be included.
Material and methods
- The methodology is well described
Results
- Results are perfectly described, it is the strong part of the manuscript.
Discussion
- You have included a paragraph on limitations, but it would be useful to describe the strengths of this study (of which there are many).
- If it has potential for application in other cancers, please add a line.
- Regarding cancer treatments add a paragraph and its association with covid infection.
Conclusion
- This is too brief, please include a few more conclusions than those expressed in the manuscript.
Author Response
Reviewer 3:
The manuscript was prepared very well. I congratulate the authors for the preparation of the manuscript. However, I have the following comments.
Introduction: Some references to the different types of cancer for which similar studies have been performed should be included.
Answer: We acknowledge the issue raised by Reviewer 3 and added some details about cancer type-specific COVID-19 study data.
Material and methods: The methodology is well described
Results: Results are perfectly described, it is the strong part of the manuscript.
Discussion: You have included a paragraph on limitations, but it would be useful to describe the strengths of this study (of which there are many). If it has potential for application in other cancers, please add a line. Regarding cancer treatments add a paragraph and its association with covid infection.
Answer: We thank Reviewer 3 for thourough assessment of our study. We added some information to the Discussion Section as suggested by Reviewer 3.
Conclusion: This is too brief, please include a few more conclusions than those expressed in the manuscript.
Answer: We added some more thoughts in the Conclusion Section, including potential applications of these data on treatment of patients with SARS-CoV-2 infections.